# I Can Step Clearly Now, the TENS Is On: Transcutaneous Electric Nerve Stimulation Decreases Sensorimotor Uncertainty during Stepping Movements

**DOI:** 10.3390/s22145442

**Published:** 2022-07-21

**Authors:** Tyler T. Whittier, Zachary D. Weller, Brett W. Fling

**Affiliations:** 1Sensorimotor Neuroimaging Laboratory, Department of Health and Exercise Science, Colorado State University, Fort Collins, CO 80523, USA; brett.fling@colostate.edu; 2Department of Statistics, Colorado State University, Fort Collins, CO 80523, USA; zdweller@rams.colostate.edu; 3Molecular, Cellular and Integrative Neurosciences Program, Colorado State University, Fort Collins, CO 80523, USA

**Keywords:** Bayesian inference, proprioception, transcutaneous electric nerve stimulation, TENS, gait, mobility, virtual reality

## Abstract

Transcutaneous electric nerve stimulation (TENS) is a method of electrical stimulation that elicits activity in sensory nerves and leads to improvements in the clinical metrics of mobility. However, the underlying perceptual mechanisms leading to this improvement are unknown. The aim of this study was to apply a Bayesian inference model to understand how TENS impacts sensorimotor uncertainty during full body stepping movements. Thirty healthy adults visited the lab on two occasions and completed a motor learning protocol in virtual reality (VR) on both visits. Participants were randomly assigned to one of three groups: TENS on first visit only (TN), TENS on second visit only (NT), or a control group where TENS was not applied on either visit (NN). Using methods of Bayesian inference, we calculated the amount of uncertainty in the participants’ center of mass (CoM) position estimates on each visit. We found that groups TN and NT decreased the amount of uncertainty in the CoM position estimates in their second visit while group NN showed no difference. The least amount of uncertainty was seen in the TN group. These results suggest that TENS reduces the amount of uncertainty in sensory information, which may be a cause for the observed benefits with TENS.

## 1. Introduction

Balance and mobility are fundamental contributors to independent living throughout the lifespan. In adults over the age of sixty-five, falls are the leading cause of fatal and non-fatal injuries [1], which is particularly troubling as is it predicted that by 2030, older adults will outnumber children for the first time in U.S. history [2]. Therefore, much research has endeavored to develop effective rehabilitative practices to mitigate, or even eliminate, mobility impairments in aging and neurodegenerative populations [3,4,5].

One therapeutic approach used to improve mobility in clinical populations is the use of transcutaneous electric nerve stimulation (TENS). TENS is a relatively new approach to improving sensorimotor function that has shown promising results and is emerging as a wearable system aimed at improving wellness and health outcomes in many populations [6,7,8]. TENS has historically been used to manage pain [9] and muscle spasticity [10], but has recently been applied to improve gait and balance in various populations [6,7,8,11]. TENS is a method of electrical stimulation in which the applied current is targeted directly at the sensory nerve fibers. When applied to improve the sensorimotor function of mobility, electrodes are placed on the muscles of the lower limbs and the applied current is set at a level below the motor threshold to minimize any evoked muscle contractions. Used in this way, action potentials in several sensory receptors are elicited both in and around the targeted muscle [12,13]. 

Recent research has found that, when applied concurrently with the clinical metrics of mobility, TENS improved the performance compared to the same metrics without TENS [6,7,8]. Almuklass and colleagues [7] applied continuous pulses just below the motor threshold in people with multiple sclerosis as well as age-matched controls as they performed metrics of sensorimotor function. The authors found that both groups (MS and controls) improved in the 6-min walk test and the MS group also improved in a timed chair rise test when compared to performing with no TENS. Additionally, Elboim-Gabyzon et al. [6] showed that patients recovering from hip surgery walked further during a 2-min walk test when receiving TENS than a group that received no stimulation. Finally, in a review article including 11 studies and 439 stroke survivors, Kwong et al. [8] concluded that TENS is beneficial to walking and mobility and improves the patients’ walking capacity.

While there is ample evidence of the benefit that TENS has on gait and mobility, the underlying mechanisms that lead to these improvements are not understood. Commonly when performing research with TENS, the amplitude is set at a level below the motor threshold (i.e., the minimal intensity of stimulation that generates an involuntary motor response) [13]. Thus, the argument that any benefit comes from the direct excitation of additional muscle activation is unlikely. Recent work has provided evidence that sensory input has a much larger impact on the overall motor function than has previously been understood [14,15,16,17,18]. Thus, the observed benefits in gait and mobility that result from the use of TENS may be due to increased sensory input relaying additional information about body orientation. However, it is not clear how the additional information is used by the central nervous system (CNS) to inform the body position awareness or construct motor plans. One hypothesis is that the additional sensory input decreases the total noise in the incoming sensory data, leading to less uncertainty in the central nervous system’s (CNS) estimation of body orientation. If this were the case, the observed benefits that accompany the use of TENS would be due to improved positional awareness, leading to more efficient motor plans. Though this hypothesis seems conceptually valid, identifying a way to measure and quantify it requires robust assessment.

Bayesian inference is a statistical model that has been used by researchers to understand how the CNS estimates body position based on uncertain sensory information [19,20]. Simply put, Bayesian inference posits that the most likely estimate of an unknown parameter comes by combining the available data with previously collected data to minimize the amount of uncertainty in the final estimate. Specific to the context of motor control, this suggests that the most certain estimate for the location of a body part is calculated by considering the incoming sensory information as well as the expected body positions based on previous movement attempts. Past research has shown that the CNS calculates the body position in a way consistent with Bayesian inference [19,21]. Additionally, we have shown in previous research that the CNS estimates the position of the center of mass (CoM) during full body stepping movements in a way that is consistent with Bayesian inference [22]. Using the Bayesian methods in a novel way, we hope to expose some of the underlying perceptual mechanisms that are benefited by the addition of TENS on the lower extremities during mobility related movements.

The purpose of this study was to clarify the underlying mechanisms that lead to the benefits of using TENS to improve gait and mobility. Using the model of Bayesian inference in motor control, we expected that the participants would display less uncertainty in their responses when estimating their CoM position during a stepping movement. Additionally, in line with previous research [22], we expected that participants would display better static balance when receiving TENS than without TENS.

## 2. Materials and Methods

### 2.1. Study Design and Set Up

Participation in this study involved two visits to our laboratory where participants completed the same protocol on each visit (Figure 1a), with the only difference between visits being whether they received TENS. A total of 31 young adults participated in this study. One participant suffered a musculoskeletal injury in between study visits and was excluded from the analysis. As a result, 30 participants were included in the final analysis. This study was approved by the Colorado State University Institutional Review Board, and all participants provided written informed consent prior to participation.

All participants were assigned to one of three study groups (accounting for balanced representations of male/females). Group NN (no TENS/no Tens) received no TENS stimulation on either of their study visits. Group NT (no TENS/TENS) received TENS stimulation on only their second visit. Group TN (TENS/no TENS) received TENS stimulation on their first visit only. All participants were healthy with no serious injuries or ailments limiting their physical abilities. A complete description of the participant and group demographics and characteristics can be found in Table 1.

Upon arrival at our laboratory, participants were fitted with the TENS electrodes on the distal and proximal end of the vastus lateralis and tibialis anterior of both legs with the cathode placed at the distal end (Figure 1b). Following similar methods of Almuklass et al. [7], the TENS intervention was applied with an FDA-approved clinical TENS device LG-TECELITE Therapy System (LGMedSupply, Cherry Hill, NJ, USA). Stimulation involved continuous asymmetrical biphasic pulses delivered with electrode pairs (2 in. × 4 in. pads) placed on the skin over the stimulated muscles. The stimulus frequency was set at 50 Hz with a pulse width of 0.2 ms. The area over the skin was shaved to minimize the electrical impedance for all participants. Electrodes were placed at the same locations on both visits, but a current was only delivered on the appropriate visit according to their group designation. For instance, participants in the NN group were fitted with the TENS electrodes on both visits although stimulation was never applied on either visit. This was conducted to eliminate any effect on performance that may be due to the additional sensory information from the TENS electrode pads. Amplitude of the TENS stimulation varied for each participant and was determined by their specific motor threshold. To identify the motor threshold for each participant, the TENS amplitude was slowly increased at 1 mA increments on each individual muscle until non-voluntary muscle contractions could either be seen or felt by the researcher. The TENS amplitude used during the assessments was 2 mA below the motor threshold for each muscle and limb [7,11]. Once the electrodes were placed, the participants completed the balance and motor control assessments. During the assessments, the TENS was only applied while the assessments were being performed and not applied in between blocks and assessments.

### 2.2. Study Assessments

For the Bayesian inference motor learning protocol, participants completed a motor learning protocol in virtual reality (VR) that involved many trials moving a cursor from a start box to a target box at different directions (Figure 2). This protocol was based on the methods of Kording and Wolpert [19] and adapted to a full body stepping task that we have previously reported [22]. In this task, the cursor position was controlled by the participants’ CoM, and they could move the cursor into the target box by taking a single step toward the target (Figure 2a). Vicon motion capture cameras continuously collected the 3-dimensional positions of a reflective marker placed on each participants’ CoM and live streamed with Vicon Tracker software (Vicon Motion System Ltd., Yarnton, UK) into the VR environment (Unity Software Inc., San Francisco, CA, USA). The CoM position was defined as fifty eight percent of each male participants’ height and fifty six percent.

For each female participants’ height measured from the ground [23]. Although the goal of the task of each trial was to place the cursor into the target box, the participants were only given visual feedback of its location briefly partway through the movement and had to estimate the final cursor’s position based on that brief feedback (Figure 2c). When they believed the cursor to be in the target box, they pressed a button with the VR controller and the final cursor position was briefly displayed to them (Figure 2d). As mentioned previously, the Bayesian model of body position estimation posits that the uncertainty present in an estimate is reduced by combining the incoming sensory information with the previously learned expectations of body position. In this protocol, the incoming sensory information is the limited visual feedback as well as continuous somatosensory and vestibular inputs. To introduce an expected position of body position, unbeknownst to participants, we added a backward radial shift to the cursor position relative to the target position on each trial (Figure 2c). With the shift added to each trial, the participants must step past the target to bring the cursor into the target box. The amount of shift varied from trial to trial, and was randomly drawn from a normal distribution N (μ = −7.5 cm, σ = 2.5 cm). No information regarding the shift was given to the participants prior to the assessment to avoid any intentional compensation in their movements. The participants first completed a training block of 100 trials with the purpose of learning the cursor shift before completing the assessment blocks. This was carried out to assist them in understanding the link between their CoM and the visual feedback of the cursor position displayed to them in VR. Following the training block, the participants completed five more blocks of 100 trials each (each block lasts roughly ten minutes). On each trial of the assessment blocks, the participants had to estimate the cursor’s position based on the limited visual feedback they received and the cursor shift that they had implicitly learnt during the training block.

To assess the static balance performance, participants completed the modified clinical test of sensory integration for balance (mCTSIB), which involves four thirty second balance trials under various sensory conditions (Figure 1b). Each trial is designed to challenge the sensorimotor system differently to identify any weaknesses an individual may have in controlling balance. For this assessment, the participants stood on a BTrackS portable force plate (Balance Tracking Systems, Inc., San Diego, CA, USA) that continuously collected the position of the participants’ center of pressure (CoP) throughout each trial. The participants stood unshod with their feet together and their hands on their hips for the duration of each trial. Trials involved standing directly on the rigid force plate with eyes open and closed as well as standing on a compliant foam pad placed directly on top of the plate with their eyes open and closed.

### 2.3. Data Analysis

For the analysis of the data of each participant from the Bayesian inference motor learning protocol, all of the responses from the assessment blocks on each visit were plotted with each trial’s cursor shift on the x-axis and the radial deviation of the cursor from the target when they pressed the VR controller on the y-axis (Figure 2e,f). In line with previous research, we used the linear relationship between these two variables for all trials within a visit to provide the outcome metrics used in our analysis [19,22,24]. In this method of analysis, the slope of the regression lines was used to measure how much the expected cursor shift influenced the body position estimates of the participants. Additionally, the amount of error from the regression line was used to represent the amount of uncertainty in the participant’s estimates for their body position. Uncertainty in a statistical view is measured by the amount of error in the estimate of a mean. To this end, we do not mean ‘uncertainty’ as the cognitive and conscious perception of the amount of uncertainty in their estimated position of their body position, but rather, we refer to the subconscious uncertainty in the CNS’s estimate for the position of the CoM that we must infer through behavioral patterns that arise through many different trials. In this way, the predicted value for Y (the value along the regression line) represents the estimate made by the CNS of the CoM position. Root mean squared error (RMSE) is a metric of the error about the regression line and therefore we used RMSE to infer the amount of uncertainty in the estimates made by the CNS of body position.

Analysis of the balance performance data was conducted using virtual time-to-contact [25,26]. Virtual time-to-contact (VTC) considers the instantaneous position, velocity, and acceleration of the center of pressure (CoP) to predict how long it would take the CoP to reach the boundary of the base of support for every data point in a trial. A lower VTC value means that it would take less time to reach the boundary of the base of support and subsequently fall, thus exemplifying a state of low stability. An average VTC value was calculated for each of the participant’s balance trials for both visits following the methods of [26]. Our hypothesis was that TENS would lead to improvements in balance performance, which would be represented by a higher VTC value with TENS.

### 2.4. Statistical Analysis

All statistical analyses were conducted in R software (version 4.1.1) with an alpha level set at 0.05. To ensure statistical assumptions were met prior to running any statistical tests, assessments of normality and equality of variance were performed on all outcome metrics of this study such as Levene’s test for equality of variance, Shapiro–Wilk tests, QQ plots, and plots of the residual vs. fitted data. These assessments indicate that all the outcome metrics used in this study met the assumptions needed and were included in our analysis.

Our first hypothesis for this study was that the participants would display less uncertainty in their estimates of their CoM position during a mobility related movement while receiving TENS. To assess the differences in uncertainty between visits and research groups, a three (groups) by two (research visits) repeated measures ANOVA was calculated with a linear mixed effects model to account for each participant being represented by more than one observation within the analyzed data. To identify specific differences between the groups and visits, follow up pairwise comparisons were calculated using Tukey’s honest significance test.

We also hypothesized that participants would display better static balance when receiving TENS than without TENS. To address this hypothesis, we analyzed the mCTSIB data from visit 1 for the NN and TN groups, who only differed by having TENS during the mCTSIB assessment. We calculated a two (groups) by four (balance condition in the mCTSIB) repeated measures ANOVA with a linear mixed effects model assessing the effects of TENS, mCSTIB condition, and any interaction effects. To identify specific differences between groups and conditions, follow up pairwise comparisons were calculated using Tukey’s honest significance test.

## 3. Results

In total, 30 neurotypical healthy adult participants were included in the final analysis. Characteristics of all of the study participants and groups are presented in Table 1. To assess the differences in the uncertainty of the participants’ responses across study visits and groups, we performed a three by two repeated measures ANOVA. The results of that ANOVA showed a main effect for visit (F_(1,207)_ = 37.28, *p* < 0.001). The corresponding means indicated that response uncertainty, on average across all participants irrespective of group, decreased from visit 1 to visit 2 (Figure 3). Additionally, we found evidence of a visit by group interaction effect (F_(2,207)_ = 3.29, *p* = 0.039). The corresponding means indicated that the change in uncertainty across visits was not the same for each group. Follow-up pairwise comparisons indicated that the groups that received TENS, whether in the first or second visit (NT and TN), both decreased in their response uncertainty from visit 1 to visit 2. Results from the post hoc comparison estimated that the uncertainty metric for the NT group in visit 1 (*M* = 2.78, *SD* = 0.57) decreased in visit 2 (*M* = 2.24, *SD* = 0.47) by 0.54 cm (*d* = 1.17, *p* < 0.001) and the TN group from visit 1 (*M* = 2.43, SD = 0.75) to visit 2 (*M* = 2.05, *SD* = 0.71) decreased by 0.39 cm (*d* = 0.84, *p* < 0.001). However, there was not enough evidence to conclude that the group that received no TENS on both visits (NN) reduced their response uncertainty from visit 1 (*M* = 2.57, *SD* = 0.59) to visit 2 (*M* = 2.40, *SD* = 0.76) (*d* = 0.36, *p* = 0.108). These results corroborate our first hypothesis that TENS would decrease the uncertainty in the participants’ estimates for where they were in space. However, since there was also a main effect of visit, we performed two additional hypothesis tests to further estimate the effect of TENS on the uncertainty metric of all groups. The first hypothesis test considered the means from all three groups at the first visit. For the first visit, two groups (NT and NN) did not receive TENS and one group (TN) received TENS. To estimate the TENS effect using visit 1 only, we subtracted the sample mean of the TN group at visit 1 from the average of the means from both groups who did not receive TENS on visit 1 (Equation (1)):(1)$$\left(\frac{{\;\overset{-}{\mu }}_{nt_{1}}+{\;\overset{-}{\mu }}_{nn_{1}}}{2}\right)-{\;\overset{-}{\mu }}_{tn_{1}},$$
where ${\;\overset{-}{\mu }}_{nt_{1}}$ represents the sample mean from group NT on visit 1, ${\;\overset{-}{\mu }}_{nn_{1}}$ represents the sample mean from group NN on visit 1; and ${\;\overset{-}{\mu }}_{tn_{1}}$ represents the sample mean from group NT on visit 1.

The estimate from computing Equation (1) was 0.24 cm. This estimate indicates that the TENS group had a lower uncertainty than the average of the non-TENS groups at visit 1. While there was not enough evidence to conclude that this effect was different from 0 (*p* = 0.263), the 95% confidence interval [−0.19, 0.67] suggests that the TENS could slightly increase the uncertainty (−0.19 cm) or provide a substantial reduction (0.67 cm). 

The second custom hypothesis test that we performed was conducted to isolate the TENS effects across visits. To do this, we compared the differences in RMSE between visits for group NT and group NN. Even though our pairwise comparison between visits for the NN group did not indicate a significant decrease in uncertainty, the results of our ANOVA showed a main effect of visit, indicating that the uncertainty value decreased from visit 1 to visit 2 regardless of group. The difference in means between visit 2 and visit 1 for the NN group represents an estimate of the visit effect on the uncertainty metric. The difference in means between visit 2 and visit 1 for the NT group represents an estimate of the visit effect plus the TENS effect. To identify the TENS effect across visits, we subtracted the difference in means in the NN group from the difference in means in the NT group (Equation (2)): (2)$$\left({\;\overset{-}{\mu }}_{{nt}_{1}}-{\;\overset{-}{\mu }}_{{nt}_{2}}\right)-\;\left({\;\overset{-}{\mu }}_{{nn}_{1}}-{\;\overset{-}{\mu }}_{{nn}_{2}}\right),$$
where ${\;\overset{-}{\mu }}_{nt_{2}}$ represents the sample mean from group NT on visit 2 and ${\;\overset{-}{\mu }}_{nn_{2}}$ represents the sample mean from group NN on visit 2.

The estimate from computing Equation (2) was 0.371 cm. This estimate suggests that the change in uncertainty due to TENS was 0.371 cm larger than the change in uncertainty due to repeated visits (0.16 cm) (CI [0.02, 0.72], *p* = 0.037). We note that the *p* values (or significance levels) from these custom hypothesis tests were not adjusted to account for multiple comparisons. 

Taken together, we believe that the results of our study provide enough evidence to conclude that TENS decreases the amount of uncertainty present in the participants’ estimates of their CoM position as they perform stepping movements.

We also hypothesized that TENS would result in participants performing better in a static test of balance. This would be measured by an increase in the VTC measurement when TENS was applied compared to when TENS was not applied. The sample means for the VTC values across the mCTSIB conditions are reported in Table 2. The results of the two by four repeated measures ANOVA with random effects indicated no significant main effect for TENS on the VTC measurement (F(1,18)= 0.320, *p* = 0.579), a significant main effect for the mCTSIB condition (F(3,54) = 73.62, *p* < 0.001), but no significant TENS by the mCTSIB condition interaction effect (F(3,54) = 0.260, *p* = 0.852). Follow-up pairwise comparisons between the mCTSIB conditions, regardless of group, all showed significant decreases from condition one to condition four (*p* < 0.001), except for conditions two and three (rigid surface eyes closed vs. compliant surface eyes open), which provided a significance level of *p* = 0.960. This decrease in VTC across mCTSIB conditions suggests a decrease in balance performance across the conditions and is in line with previous research [27].

## 4. Discussion

The purpose of this study was to clarify the underlying mechanisms that lead to the observed improvements in gait and balance that are seen with the use of TENS. To address this purpose, we applied a theoretical model of Bayesian inference to measure and assess sensory uncertainty and how it influences the body position estimates during a full body stepping movement. Furthermore, we applied a crossover study design that consisted of three study groups performing the Bayesian motor learning assessment on two separate visits either with or without the addition of concurrent TENS. The results of our analyses are in line with our hypothesis that TENS decreases the uncertainty that participants showed as they estimated their body position in our assessment. The following section will elaborate on the main findings gathered from this study and discuss future applications for how this knowledge can be used to benefit this field of research.

Taking all of the results from our analyses in this study together, we concluded that the amount of uncertainty in the participants’ estimations of their CoM position during a stepping movement was decreased with the addition of TENS. This main finding provides insights into the perceptual mechanisms that are affected by TENS and leads to the observed improvements in motor performance with its application [7,28]. With less sensorimotor uncertainty regarding the position of the involved body parts, the CNS can make more efficient motor and plans to execute the movements involved in goal-oriented motor skills. For example, when navigating through a crowded grocery store, the CNS has a clearer estimate for the foot’s position when forced to avoid tripping on a fellow shopper’s cart. This increased clarity may also lead to improvements in movement efficacy in populations known to have sensory and motor impairments due to disease or injury such as multiple sclerosis and diabetes [7,28]. The low cost of a TENS unit makes this a simple form of therapeutic intervention that may have a serious benefit to multiple populations. Future research will continue to investigate how improvements in sensorimotor uncertainty can benefit motor performance in various populations as well as the best practices of applying this technology to maximize its effects.

Furthermore, we showed that group TN, who received TENS on their first visit only, demonstrated the least response uncertainty in their first visit when compared to other groups on the same visit. Interestingly, group TN went on to further decrease their response uncertainty in the second visit without TENS and still reported less uncertainty than both groups on the second visit. Figure 4 shows the individual data from a participant in both groups who received TENS and exemplifies the advantage of receiving TENS on the first visit and the continued benefit after 2 weeks. A constant concern that is associated with many forms of sensory augmentation is whether any observed benefits will persist once the additional sensory stimulus is removed. One implication of our findings is that, at least in certain circumstances, the benefit that is gained from the addition of TENS while learning and performing a new movement is retained in future performances of the same movement. Furthermore, when compared to group NT, who received TENS on their second visit, it seems that TENS is most beneficial when it is applied early in the motor learning process. Recent work has shown that when learning a new motor skill, functional changes occur in the somatosensory cortex to process incoming sensory data prior to any observed changes in the specific motor areas of the brain [16,29]. Taken together, this further emphasizes the importance of sensory input to motor performance and specifically to motor learning. As mentioned previously, Bayesian motor control theory posits that the CNS combines incoming sensory information with learned expectations of body position based on previous attempts.

However, when performing a novel movement, expectations of body position are often ill-informed or absent altogether. In this case, learning the physical and sensory consequences of a new motor skill is a priority to ensure accurate position estimation and movement performance. From this perspective, it seems logical that enhancing the incoming sensory information with TENS would assist the CNS in identifying pertinent sensory information that informs it of bodily states while performing new movements. The finding that this benefit is retained and even continues to improve following a two-week washout period is compelling and merits the need for further examination. An intervention as simple and cost-effective as TENS could be incredibly beneficial to clinical populations striving to learn, or re-learn, new motor skills in response to injury or disease.

Interestingly, we found that the addition of TENS had no effect on static balance, as assessed by the mCTSIB. Much previous research has found external electric stimulation of the lower limbs to be effective at improving balance metrics [13,30,31]. However, Paillard et al. [32] recently found that the participants’ responsiveness to the electrical stimulation of sensory nerves to improve balance depends on their baseline balance abilities. Our study included only young participants that historically have exceptional balance abilities. It is possible that any effect of TENS on the balance performance in this healthy population goes unnoticed because they are already proficient at controlling balance. Additionally, it is of interest that much of the previous research that has shown improvements in performance with the addition of TENS included methods of assessment that were more dynamic in nature, requiring the movement of many joints in multiple planes of motion, in contrast to the static conditions inherent to our balance assessment [7,8].

With the findings from this study, we can conclude that the addition of TENS to the muscles of the lower extremities leads to a decrease in sensorimotor uncertainty for the position of the CoM during a stepping movement. This conclusion is based on the results of a repeated measures ANOVA that we performed as well as the follow-up pairwise comparisons. When interpreting these results, the reader must bear in mind a few key principles. First, it needs reminding that the uncertainty metric that we used as our outcome metric is from the RMSE of a regression line of each participant’s response in their study visits. The RMSE is meant to represent the variability in the estimate made by the CNS of the CoM position throughout a stepping movement based on the incoming sensory information as well as the learned expectations for the CoM position. We acknowledge that the true metric of uncertainty for each participant is a latent variable that cannot be directly observed, but rather, must be inferred through carefully designed methods. We are seeking to measure a perception that is built upon sensory information and demonstrated through skilled behavioral patterns that arise through many trials. Though novel and unique, we are confident in the results of this study and the conclusions drawn from them about how sensory information can be manipulated to improve performance.

Next, we based our main conclusions on the results of the four statistical tests that we performed. The results of the ANOVA that we calculated indicated that the three groups responded significantly different in their uncertainty measurement across visits 1 and 2. We also performed Tukey’s honest significance tests, which revealed that both groups who received TENS reduced their uncertainty values from visit 1 and visit 2, whereas the NN group did not. However, because there was evidence of an overall visit effect in the ANOVA (*p* < 0.001), it is possible that the number of trials and participants included in this study were not enough to see a significant difference in this metric. Through further follow-up analyses on subgroups in our study, we performed custom hypothesis tests designed to isolate the TENS effect on uncertainty. We showed that the TENS led to a decrease in uncertainty for the group who received TENS on the first visit, although this was not a significant decrease. However, with the reported confidence interval, we can be 95% confident that the true change in uncertainty was between +0.19 cm (an increase) and −0.67 cm (a decrease) because of TENS. In isolation, this would not be sufficient data to conclude that TENS decreases uncertainty, however, it has sparked interest into further analyses to parse out the TENS effect. To this end, we performed an additional hypothesis test seeking to estimate the TENS effect across visits. We found that TENS led to a decrease in uncertainty from visit 1 to visit 2. As mentioned previously, we did not adjust for multiple comparisons, which increases our chances of a Type I error. However, as this analysis was exploratory in nature, we feel confident that taking all of these results in total, we can conclude that the addition of TENS to the muscles of the lower limbs leads to a decrease in sensorimotor uncertainty for the position of the CoM during a stepping movement. We invite future research to be performed to further identify the effects of TENS on sensory uncertainty.

The exact mechanisms that lead to the decrease in positional uncertainty due to TENS remain to be seen. Previous work using various methods of peripheral stimulation of sensory fibers have suggested stochastic resonance may be a large reason for the benefits that accompany these types of sensory augmentation [13,33]. In this sense, the electrical stimulation provided by TENS may add low-level noise that enhances the detection and transmission of weak sensory signals by amplifying the total signal and, as a result, the sensory cues most important to coordinating the current motor task. Furthermore, Paillard [32] suggests that this also can change the ion permeability of the mechanoreceptors (group Ia and IIa afferents of muscle spindles), priming them, in a sense, to make them more likely to fire action potentials and increase sensory input to the brain and spinal cord. Applying a similar study purpose to decrease sensory uncertainty, Macerello and colleagues applied peripheral nerve stimulation with high-frequency vibration to the muscle of the wrist as healthy and clinical participants completed a battery of upper extremity motor tasks [34]. They found that both groups decreased the completion time of the motor tasks and showed a decrease in the EEG beta power over the sensorimotor cortices as they received the stimulation. Altogether, our results combined with previous research, support the hypothesis that TENS, and other methods of afferent stimulation, improve motor performance by decreasing the noise inherent to sensory data and permitting users to be more certain of their body position as they perform motor tasks.

## 5. Conclusions

In conclusion, we demonstrated that TENS applied to the muscles of the lower extremities while performing a multi-directional full body stepping motion decreases the uncertainty in sensory information and improves the participants’ estimation of the location of their CoM. Furthermore, we demonstrated that the Bayesian model of sensorimotor uncertainty can be used to assess and measure the underlying processes that benefit from a therapeutic device aimed at improving sensory function. Future work applying these findings and methods to various contexts is needed to further understand the underlying mechanisms that enable effective gait and mobility in all populations.

## Figures and Tables

**Figure 1 sensors-22-05442-f001:**
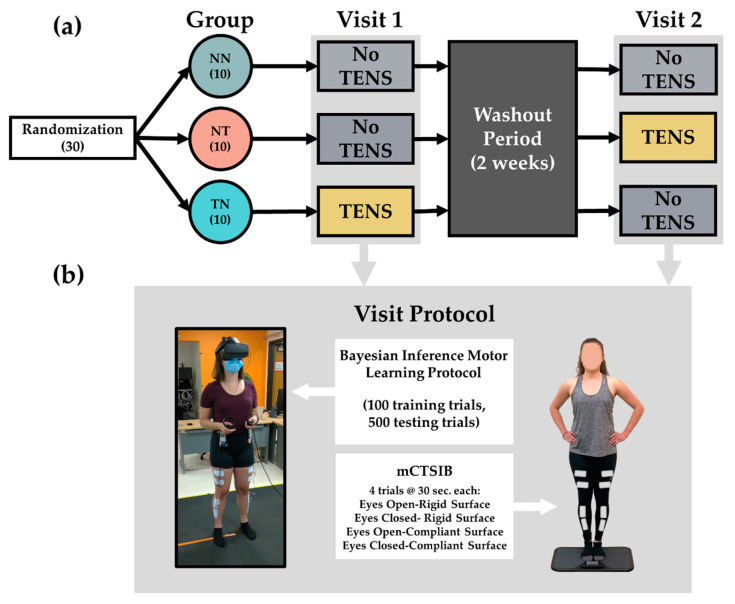
Study design and protocol. (**a**) Thirty healthy young participants were pseudo-randomly assigned to one of three study groups. The NN group received no TENS on either visit. The NT group received TENS only on their second visit. The TN group received TENS on their first visit. (**b**) Visit protocol involved the completion of a motor learning protocol performed in virtual reality as well as the completion of the modified clinical test of sensory integration for balance (mCTSIB).

**Figure 2 sensors-22-05442-f002:**
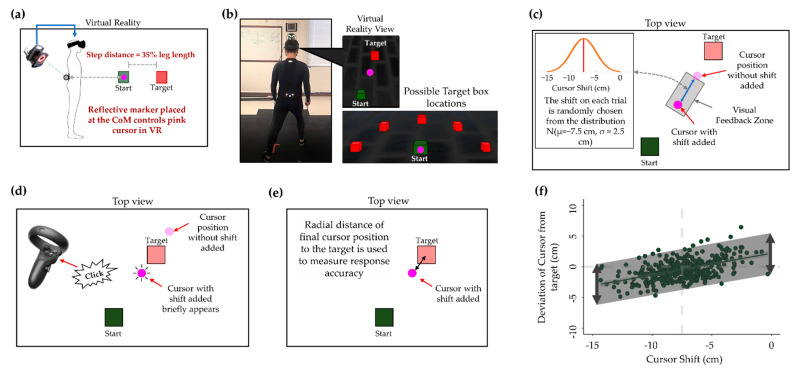
The Bayesian inference motor learning protocol and analysis. Participants completed a training block of 100 trials to learn to expect the radial shift added to the cursor position, and then five more blocks of 100 trials that were included in the final analysis. CoM = center of mass. (**a**) Participants bring a cursor (controlled by their CoM) from a start box to a target box with a single step. (**b**) On each trial, the target will appear in one of five positions, prompting a step in different directions, depending on the target displayed. (**c**) Participants only get to see the cursor within the visual feedback zone where a backwards radial shift is added to its position, prompting them to step further than necessary to bring the cursor into the target position. (**d**) Participants click a button on the VR controller when they believe the unseen cursor is in the target box, after which its position is briefly shown to promote continuous learning of the backwards shift. (**e**) The amount of cursor shift and radial distance of the final cursor position from the target on each trial are used for analysis. (**f**) The root mean squared error (RMSE) of the regression line between the amount of cursor shift was used to quantify the amount of uncertainty in participants’ responses for each study visit.

**Figure 3 sensors-22-05442-f003:**
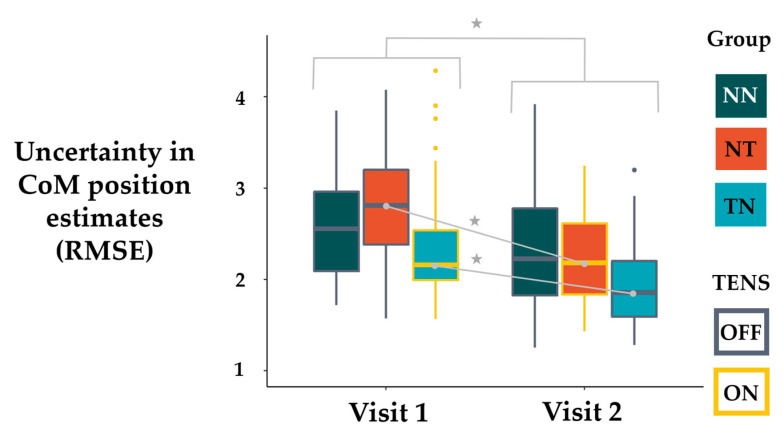
The response uncertainty for the three groups across visits. The groups that received TENS significantly decreased their response uncertainty from visit 1 to visit 2. ★ = *p* value < 0.001.

**Figure 4 sensors-22-05442-f004:**
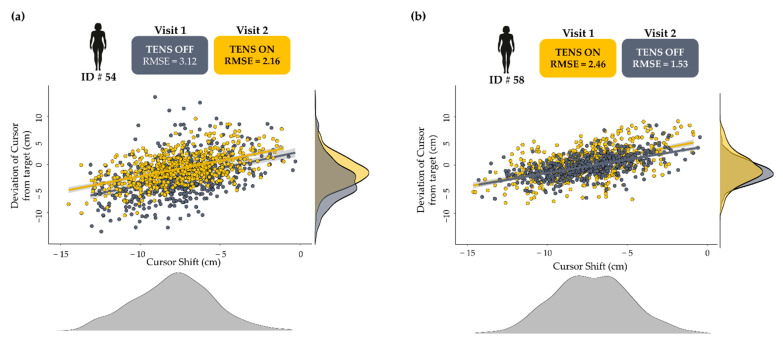
The scatter and density plots of the two participants’ responses who received TENS on their second and first visit, respectively. We found that both groups reduced the uncertainty in their responses from visit 1 to visit 2, with the group who received TENS on their first visit (TN) reporting the least degree of uncertainty. (**a**) Data from a participant that received TENS on the second visit. (**b**) Data from a participant that received TENS on the first visit. Both participants reduced the uncertainty in their estimates from visit 1 to visit 2 with the participant in (**b**) showing the lowest uncertainty.

**Table 1 sensors-22-05442-t001:** The group demographics and characteristics.

Group	NN	NT	TN	*p*-Value ^1^
n	10	10	10	
Sex: Male (%)	4 (40.0)	4 (40.0)	5 (50.0)	0.87
Age: Years (mean (SD))	23.40 (2.2)	24.70 (4.0)	23.80 (3.4)	0.67
Height: Meters (mean (SD))	1.72 (0.1)	1.72 (0.1)	1.74 (0.1)	0.86
Weight: Kilograms (mean (SD))	74.30 (17.1)	73.1 (16.0)	77.7 (14.1)	0.79
BMI: kg/m^2^ (mean (SD))	24.96 (4.0)	24.41 (3.5)	25.64 (4.4)	0.79
Exercise: Min/week (mean (SD))	318.89 (202.0)	394.00 (219.9)	343.75 (103.5)	0.66

NN = No TENS/No TENS, NT = No TENS/TENS, TN = TENS/No TENS. ^1^
*p* values represent the significance level from a group effect in a one-way ANOVA with each respective variable as the response variable.

**Table 2 sensors-22-05442-t002:** The VTC results for mCTSIB balance conditions with and without TENS.

	mCTSIB Condition			
	Rigid Surface Eyes Open	Rigid Surface Eyes Closed	Compliant Surface Eyes Open	Compliant Surface Eyes Closed
TENS OFF	1.32(0.22)	1.15(0.21)	1.13(0.23)	0.75(0.08)
TENS ON	1.38(0.30)	1.16(0.22)	1.21(0.23)	0.78(0.16)

VTC = virtual time-to-contact. Reported in seconds, where larger values represent better balance when compared to smaller values. Values reported as mean(sd).

## Data Availability

Not applicable.

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
