# Peer review of "I Can Step Clearly Now, the TENS Is On: Transcutaneous Electric Nerve Stimulation Decreases Sensorimotor Uncertainty during Stepping Movements"

_sensors, 2022, doi:10.3390/s22145442_

Round 1

Reviewer 1 Report

The current manuscript under review contains data collected from 30 healthy subjects assessing the TENS impact on uncertainty during stepping. The authors utilized data collected from 3-D motion tracker and VR environment and Bayesian model to quantify the response accuracy of a cursor's position on target task. CTSB  test was utilized to assess the static balance performance. The noted results in the manuscript indicate that TENS improves the participant's estimation about body orientation.

The concept behind this study is very interesting and important, the authors point out that little is know regarding the mechanism that TENS improves the perception during mobility and gait. Additionally, given the variety of sensorimotor  compromised patient population, these results are potentially meaningful for healthcare providers. 

However, while these results are interesting there are several considerations and questions that should be addressed.

1. While the title of the manuscript indicates TENS helps stepping clearly, it is unclear whether the performed performed a stepping task. For instance, it seems as if the participants performed one single step towards the target. Accordingly, it would be worth considering a revision to the title.

2. The application of TENS electrodes on NN is unclear and needs more description. For instance, placing the electrodes and not turning on the device would benefit to blind the participants to their group. Also, think about the additional sensory information that the pad would provide to the skin.

 3. The utilization of CTSIB protocol is unclear and needs more description. The authors state that CTSIB was recorded under 4 conditions however results lack the statistical report for this test. From the images presented in the manuscript, it seems participants wore no shoes, please provide more description for clarity. Also, visit Clinical Test of Sensory Interaction on Balance | RehabMeasures Database (sralab.org) for comparing your results with reports available among healthy groups.

Reviewer 2 Report

Thank you for the invitation to review the manuscript entitled ‘I can step clearly now, the TENS is on: Transcutaneous electric nerve stimulation decreases sensorimotor uncertainty during stepping movements. This study aims to evaluate the influence of TENS on the motor learning process of a stepping task that adopts the Bayesian model. Results show that TENS reduce the uncertainty of CoM estimation, ie the RMSE of the linear regression of expected cursor shift and participants’ body position estimates. Results of the study showed that participants are more certain about their position when TENS is applied. Besides, applying TENS in the first session seems to induce a prolonged learning effect or facilitated motor learning.

In general, the manuscript is well-written and addresses and provides preliminary evidence that, on top of inducing an immediate effect on the sensorimotor system, this modality also facilitates motor learning.

The authors have reviewed the relevant literature. Primary findings are clearly presented and discussed. The conclusion is appropriate and supported by the results.

My only major concern is that

1.          The author may need to describe in the background session how this study fits into the scope of ‘Sensors’ or the special issue.

Below are other comments/ suggestions for consideration.

 2.       Besides,please check with the editorial office whether consent from the lady shown in figure 1b is needed (if the authors haven’t submitted it previously). Since her face can be seen clearly.

3.          Were the participants told that there was a backward cursor shift in the visual feedback zone? I hope the authors can describe this in the manuscript.

4.          What are the inclusion and exclusion criteria of participants?

5.          Please try to justify the sample size.

6.          The word ‘Randomization’ seems inappropriate to me, since patients are stratified based on gender as described in the text.

7.          What was the approximate duration of each assessment block?

8.          Please provide the effect size estimates for the post-hoc comparison for the uncertainty measures (Line 262-266).

9.          Line 265 ‘wasn’t’ please avoid contraction

10.      Although TENS had no effect on the VTC score, I think the author should at least provide the mean +/- SD of this score for information.

11.      Table 1: please use SI unit (m and kg) instead of inches and pounds
